# Effect of Probiotic Therapy on Neuropsychiatric Manifestations in Children with Multiple Neurotransmitter Disorders: A Study

**DOI:** 10.3390/biomedicines11102643

**Published:** 2023-09-26

**Authors:** Loredana Matiș, Bogdana Ariana Alexandru, Radu Fodor, Lucia Georgeta Daina, Timea Claudia Ghitea, Silviu Vlad

**Affiliations:** 1Faculty of Medicine and Pharmacy, Medicine Department, University of Oradea, 410068 Oradea, Romania; matisloredana@yahoo.com (L.M.); ariana.bogdana@gmail.com (B.A.A.); dr.radu.fodor@gmail.com (R.F.); lucidaina@gmail.com (L.G.D.);; 2Faculty of Medicine and Pharmacy, Pharmacy Department, University of Oradea, 410068 Oradea, Romania

**Keywords:** neurotransmitter, gastrointestinal disorders, neuropsychological imbalance, psychobiotics

## Abstract

Probiotics, also known as psychobiotics, have been linked to cognitive functions, memory, learning, and behavior, in addition to their positive effects on the digestive tract. The purpose of this study is to examine the psychoemotional effects and cognitive functioning in children with gastrointestinal disorders who undergo psychobiotherapy. A total of 135 participants, aged 5–18 years, were divided into three groups based on the pediatrician’s diagnosis: Group I (Control) consisted of 37 patients (27.4%), Group II included 65 patients (48.1%) with psychoanxiety disorders, and Group III comprised 33 individuals (24.4%) with psychiatric disorders. The study monitored neurotransmitter levels such as serotonin, GABA, glutamate, cortisol, and DHEA, as well as neuropsychiatric symptoms including headaches, fatigue, mood swings, hyperactivity, aggressiveness, sleep disorders, and lack of concentration in patients who had gastrointestinal issues such as constipation, diarrhea, and other gastrointestinal problems. The results indicate that psychobiotics have a significant impact on reducing hyperactivity and aggression, and improving concentration. While further extensive studies are needed, these findings offer promising insights into the complexity of a child’s neuropsychic behavior and the potential for balancing certain behaviors through psychobiotics.

## 1. Introduction

Serotonin is primarily responsible for regulating our emotions. Acting in conjunction with adrenaline and dopamine, it elevates mood and controls motivation. It also has a relaxing effect, improves sleep, and acts as an antidepressant. Furthermore, it plays a role in regulating satiety and pain sensitivity. Serotonin also influences essential gut functions and nutrient absorption through the gut [1].

Initially, an intermediate substance called 5-hydroxytryptophan (5-HTP) is produced from the amino acid tryptophan [2]. Serotonin is then produced in a second step, which requires vitamin B6. The “sleep hormone” melatonin is formed from serotonin. Therefore, a serotonin deficiency can lead to melatonin deficiency and, consequently, severe sleep difficulties [3].

The fundamental component of serotonin, tryptophan, is obtained from various foods. Foods particularly rich in tryptophan include soybeans, mung beans, peanuts, cashews, sunflower seeds, some types of cheese (such as parmesan, Emmental, edam, brie, camembert, or Gruyère), eggs, meat, fish (especially tuna, salmon, mackerel, and trout), oatmeal, and wheat germs [4,5].

Significant amounts of vitamin B6 can be found in the following foods: whole grain products, potatoes, bananas, legumes (such as soybeans and lentils) [6], avocados, carrots, Brussels sprouts, sunflower seeds, nuts, liver, meat, and fish [7].

In the context of stress, GABA (gamma-aminobutyric acid) plays a major role. It has a calming effect and influences the levels of stress hormones [8]. GABA is crucial to memory and learning. It prevents sensory overload, has an anxiolytic and relaxing effect, promotes sleep, reduces pain, possesses antispasmodic properties, and stabilizes blood pressure [8]. GABA is produced from glutamate and requires vitamin B6 for synthesis [9].

Many individuals are sensitive to foods containing glutamate [10]. However, contrary to popular belief, this sensitivity does not seem to be the cause of the so-called “Chinese restaurant syndrome”, which can result in nausea, stomach pain, headache, and diarrhea after dining at a Chinese restaurant [11]. It is important to note that glutamate ingested through food does not reach the brain [12].

Therefore, in the following discussion, we do not refer to glutamate in food but rather to the neurotransmitter in the brain, composed of glucose and glutamic acid [13]. Glutamate serves various crucial functions in the brain [14]. As a neurotransmitter, it is essential to complex brain tasks, such as learning and memory, purposeful and controlled movements, and the brain’s ability to perceive, and adapt to, the environment [15]. Additionally, glutamate participates in metabolic processes that detoxify the brain from harmful ammonia [16].

Cortisol has diverse effects, including increasing the metabolism and blood sugar for energy production, regulating fat distribution in the body (with fat accumulating in the abdominal region), increasing appetite, reducing pain sensitivity, altering emotional sensitivity, inhibiting growth processes, lowering immunity, and inhibiting inflammation. However, a continuous high level of cortisol can increase inflammatory activity in the body [17,18].

Cortisol production from cholesterol occurs in the adrenal glands’ cortex and follows a circadian rhythm [19]. During sleep, in the second half of the night, the body produces the largest amount of cortisol, resulting in the highest cortisol levels immediately upon waking in the morning. Cortisol levels decrease rapidly in the early afternoon and are gradually declining by late evening. In the second half of the night, cortisol levels rise again significantly. In this basic rhythm, cortisol levels also rise slightly and rapidly in response to acute stress during the day [20,21].

A low level of DHEA (dehydroepiandrosterone) is associated with various degenerative processes in the body, which means that DHEA levels can serve as an estimate of a person’s biological age. Additionally, DHEA serves as the precursor of sex hormones such as testosterone and estrogen [22].

DHEA has a short half-life of approximately 10–15 min. Consequently, it is mainly found in the form of the DHEA-S depot, which can be converted into DHEA as needed [23]. DHEA acts as a direct regulator of cortisol and helps balance the stress response induced by cortisol, thereby contributing to stress management [24]. It also has a beneficial effect on muscle building and increases the level of HDL cholesterol, known as “good cholesterol”, which helps to reduce fat deposits in blood vessels, thus preventing atherosclerosis. DHEA also possesses anti-inflammatory properties and enhances the immune system [25].

This paper aims to assess the impact of changes in neurotransmitter levels on the mental health status of children following an extended adjustment process [26]. Prebiotics have been shown to benefit mental health and support the development of specific commensal bacteria with effects on mental health. Therefore, they can be considered part of the definition of psychobiotics [27,28,29]. In this context, the most extensively studied prebiotic substrates with neural effects are, primarily, FOS and GOS, as they promote the beneficial growth of *Bifidobacteria* and *Lactobacilli* [30].

The effects of psychobiotics extend beyond the regulation of the neuroimmune axes, such as the hypothalamic–pituitary–adrenal (HPA) axis, the medullary sympathoadrenergic axis (SAM), and the inflammatory response, in nervous system disorders. They are also associated with cognitive function, memory, learning, and behavior.

One of the primary objectives of this research is to evaluate the impact of these changes on the mental and physical health status of children, with a particular focus on psychoemotional effects and cognitive functioning following a specific probiotic therapy (psychobrobiotherapy). Additionally, another crucial objective is to determine the essential role of the adjustment of neurotransmitter levels in the context of pediatric medical practice, in order to minimize the risks of mental health impairment and maximize the future development of children.

## 2. Materials and Methods

The aim of this study was to investigate patients treated at a private nutrition practice in Oradea, Romania, from 2020 to 2022, in accordance with the World Medical Association Declaration of Helsinki guidelines. The selected patients, aged from 5 to 18, had gastrointestinal disorders such as non-infectious diarrhea and constipation, and other gastrointestinal disorders, such as flatulence, feeling full, gas, belching, and abdominal pain.

Of the 1145 patients initially evaluated, 135 were chosen to participate in the study and underwent monthly consultations in adherence with the same guidelines. The exclusion criteria were an age less than 5 years or greater than 18, drug treatment that could alter neurotransmitter concentrations (SSRIs, SNRIs), and refusal to take part in the study. The primary objective of the study was to analyze the correlation between oxidative stress and clinical and paraclinical parameters, as well as to identify differences between the study groups.

To ensure sample consistency, individuals over 18 years of age, those who declined to participate, and those with chronic diseases that could potentially influence the results were excluded. The sample size was determined using a formula specific to such studies, resulting in a required number of 85 cases to achieve a 95% probability level.

All patients had gastrointestinal disorders and adhered to healthy eating recommendations, which involved a controlled caloric intake and set meal times. In the two study groups, they also received a personalized probiotic treatment tailored to address their specific gastrointestinal issues. The recommended probiotics varied in terms of proportions and combinations, and they included *Bifidobacteria*, *Lactobacilli*, and *Saccharomyces* ssp., with formulations that excluded gluten, dairy, yeast, or egg. The dose of probiotics varied according to age and weight. In patients with constipation, it was supplemented with inulin.

These individuals had imbalances and received natural treatment (Melissa extract, rhodiola, magnesium, vitamin B6) without SSRI or SNRI treatment in order to assess the effectiveness of probiotic therapy and diet therapy.

### 2.1. Clinical Analysis

The clinical assessment was conducted in a medical office, with a focus on general symptoms such as headaches, fatigue, mood swings, hyperactivity, aggression, sleep disorders, and lack of concentration. The patient’s medical history included the identification of any personal pathological history, medication use, smoking, alcohol consumption, or other prohibited substances.

### 2.2. Paraclinical Analysis

Paraclinical evaluations were carried out to support the diagnosis. Paraclinical analyses of neurotransmitter levels, including serotonin, GABA, glutamate, cortisol, and DHEA, were conducted at the beginning and end of the research period, using enzymatic, colorimetric, and spectrophotometric methods, in the analysis laboratory. Specific urine and saliva tests were also employed to assess the presence of stress hormones in the body (CTL and Ortholabor GmbH, 26160 Bad Zwischenahn, Germany).

### 2.3. Statistical Analysis

The statistical analysis involved an investigation of biomarker changes over time. Numerical and graphical summaries of case profiles, including changes relative to the baseline values, were provided. The biomarker distributions showed no significant deviation from normality. A linear mixed model with random effects and an unstructured correlation for repeated measures was utilized to model changes over time. The time of testing was incorporated as a fixed effect, initially as a categorical variable for comparing mean changes and, subsequently, as a continuous variable for analyzing biomarker time trends. Pearson correlations were used to assess the relationships between biomarkers.

The analyses were performed using SPSS software (IBM, Chicago, IL, USA, version 20), with the level of significance set at *p* < 0.05. Statistical tests included the ANOVA test for two variables and the post hoc Bonferroni test for three variables. The residuals of the fitted model were examined to assess its fit for each biomarker at each time point.

## 3. Results

Demographic Overview:

Of the 135 patients included in the study, 54 were male, representing 40.0% of the total, while 81 were female, constituting 60.0% of the total. The mean age of these participants was 12.57 years, with a standard deviation of 4.43. The minimum age observed was 5 years, while the maximum age was 18 years. It was noted that the majority of patients came from urban environments, accounting for 60% of the total. Statistically, the research groups exhibited normal distributions, as confirmed via the skewness and kurtosis test, with values falling within the range from −3.00 to +3.00. The participants were categorized, according to the pediatrician’s diagnosis, into three groups: Group I (the control) comprised 37 patients (27.4%), Group II consisted of 65 patients (48.1%) with psychoanxiety disorders, and Group III included 33 individuals (24.4%) with psychiatric disorders.

### 3.1. Neurotransmitter Levels

#### 3.1.1. Serotonin

Serotonin, often referred to as the “happiness hormone”, is a crucial neurotransmitter in the brain, present in both the central nervous system and the intestinal lining. It plays a role in regulating satiety and pain sensitivity. The normal values for serotonin fall within the range of 100–225 µg/g creatinine. Consequently, among the 135 individuals analyzed, it was observed that 54 people (40%) had values within the normal range (100–225 µg/g creatinine), while 54 people (40.0%) exhibited values exceeding 225 µg/g creatinine. Serotonin levels below the normal limits were recorded in 27 people. The distribution of the recorded serotonin values is illustrated in Figure 1.

The initial statistical differences between groups, as indicated by the ANOVA coefficient (F), were significant in the case of serotonin (F = 25.468, *p* = 0.001). Statistically significant differences (*p* < 0.01) were also observed in comparisons between group I and group II, as well as between group II and group III, while statistically insignificant differences were found between group I and group III. However, the dopamine levels did not differ significantly between patients with psychoanxiety disorders and those with psychiatric disorders.

The data pertaining to the distribution of dopamine within the research groups are depicted in Figure 2.

#### 3.1.2. GABA

GABA and glutamate are two exceptionally important neurotransmitters in the brain. These neurotransmitters’ formation processes are closely interrelated and dependent on each other. They function as direct opposites, jointly regulating the level of brain activity. The normal values for these neurotransmitters fall within the range of 2.25–12.8 µmol/g creatinine. Consequently, among the 135 individuals analyzed, none of them recorded values below the normal threshold (<2.25 µmol/g creatinine), while 81 people (60.0%) had values within the normal range (2.25–12.8 µmol/g creatinine), and 27 people (20.0%) exhibited high values exceeding 12.8 µmol/g creatinine. The distribution of the recorded GABA values is depicted in Figure 3.

The initial statistical differences between groups, as indicated by the ANOVA coefficient (F), were significant in the case of GABA (F = 30.944, *p* = 0.001). There were no statistically significant differences (*p* > 0.01) regarding the distinctions between group I and group III, but statistically significant differences were observed between group I and group II (*p* < 0.05), as well as between group II and group III (*p* < 0.05). The level of GABA significantly varied between patients with psychoanxious disorders and those with psychiatric disorders, with group 2 exhibiting a greater excess.

The data concerning the distribution of GABA among the research groups are illustrated in Figure 4.

#### 3.1.3. Glutamate

Glutamate is present in many foods and contributes to their pleasant taste. It is also recognized as a flavor enhancer in commercially prepared foods and spices, where it is used in doses many times higher than in natural foods. It may also be listed in the ingredients as “monosodium glutamate” or “E621”, for example. The typically accepted values range between 8 and 30 µmol/g creatinine. Consequently, among the 135 individuals analyzed, it was found that 81 people (60%) had values below the normal range, while 54 people (40.0%) were registered with values exceeding 30 µmol/g creatinine. The distribution of the recorded glutamate values is depicted in Figure 5.

The initial statistical differences between groups, as indicated by the ANOVA coefficient (F), were significant in the case of glutamate (F = 69.650, *p* = 0.001). Statistically significant differences (*p* < 0.01) were also observed concerning the distinctions between group I and group II, as well as between group II and group III. Additionally, statistically significant differences (*p* < 0.01) were noted between group I and group III. The level of glutamate significantly varied between patients with psychoanxiety disorders and those with psychiatric disorders, with the control group showing no excess in glutamate levels.

The data regarding the distribution of glutamate among the research groups are illustrated in Figure 6.

#### 3.1.4. Cortisol

Cortisol, also referred to as the “stress hormone”, typically influences the body’s response to stimuli and is secreted in reaction to stress. It plays a vital role in the intricate regulation of other neurotransmitters involved in stress management. The normal values for morning cortisol levels range between 4 and 12 ng/mL creatinine. Thus, among the 135 individuals analyzed, it was observed that 81 people (60%) had values below the normal range, while 54 people (40.0%) registered values exceeding 12 ng/mL creatinine.

Normal midday cortisol values fall within the range of 1.5–5 ng/mL creatinine. Consequently, among the 135 individuals analyzed, it was found that 108 people (80%) had values below the normal range, while 27 people (20.0%) exhibited creatinine values exceeding 5 ng/mL.

Normal evening cortisol values range from 0.5 to 1.8 ng/mL creatinine. Among the 135 individuals analyzed, it was identified that 108 people (80%) had values below the normal range, whereas 27 people (20.0%) had creatinine values exceeding 1.8 ng/mL.

The distribution of the morning, noon, and evening cortisol values is illustrated in Figure 7.

The initial statistical differences between groups, as indicated by the ANOVA coefficient (F), were significant in the case of morning cortisol (F = 65.182, *p* = 0.001). Statistically significant differences were also observed (*p* < 0.01) between group I and group II, as well as between group I and group III, and between group II and group III. The morning cortisol level was significantly higher in individuals with psychiatric disorders.

Morning cortisol levels below 4 ng/mL were not recorded in any individual. Levels between 4 and 12 ng/mL were observed in 10 people from group I, 65 from group II, and 6 from group III. Cortisol levels exceeding 12 ng/mL were recorded in 27 individuals from groups I and III.

Afternoon cortisol levels below 1.5 ng/mL were not observed in any of the participants. Levels between 1.5 and 5 ng/mL were recorded in 10 people from group I, 65 people from group II, and 33 people from group III. Cortisol concentrations at lunch exceeding 5 ng/mL were recorded in 27 individuals from group I.

Evening cortisol levels below 0.5 ng/mL were not observed in any participant. Levels between 0.5 and 1.8 ng/mL were observed in 10 people from group I, 65 people from group II, and 33 people from group III. Cortisol levels exceeding 1.8 ng/mL in the evening were observed in 27 patients from group I. A graphical representation can be seen in Figure 8.

#### 3.1.5. DHEA

The hormone DHEA is primarily produced from cholesterol in the adrenal gland. DHEA production gradually declines with age, typically beginning at age 25. A low level of DHEA is associated with several degenerative processes in the body. The normal values for morning DHEA levels range between 83 and 496 pg/mL creatinine. Among the 135 individuals analyzed, it was observed that 54 people (40%) had values below the normal range, and 54 people (40.0%) registered values exceeding 496 pg/mL creatinine, with only 27 people having values within the normal range.

The normal values for evening DHEA levels fall within the range of 36–216 pg/mL creatinine. Consequently, out of the 135 individuals analyzed, 81 people (60%) had values within the normal range, 27 people had values below these norms (<36 pg/mL), and 27 people (20.0%) recorded values exceeding 216 pg/mL creatinine.

The distribution of the recorded morning and evening DHEA values is presented in Figure 9.

The initial statistical differences between the groups, as indicated by the ANOVA coefficient (F), were significant in the case of the morning DHEA (F = 32.894, *p* = 0.001). Statistically significant differences (*p* < 0.01) were also observed between group I and group II, as well as between group I and group III, and between group II and group III. The morning DHEA level was significantly higher in individuals with psychoanxiety disorders, and the lowest level was recorded in group 3 (comprising individuals with psychiatric disorders), and was statistically significant in several cases.

The data regarding the distribution of the morning DHEA among the research groups are presented in Figure 10.

### 3.2. Neuropsychiatric Manifestations in Patients with Gastrointestinal Disorders

Significant differences were observed in the cases of headache, fatigue, mood swings, hyperactivity, aggression, sleep disturbances, and initial lack of concentration among patients in the three research groups. By the end of the research period, there was a significant improvement in hyperactivity, aggression, and lack of concentration, while the other parameters showed improvement but did not reach the threshold of statistical significance. In terms of neuropsychological parameters, significant differences (*p* < 0.01) were noted for mood swings, hyperactivity, and lack of concentration among the three research groups, as presented in Table 1.

### 3.3. Correlations

From the Pearson correlation analysis, a significant and directly proportional relationship was observed between the differences in headache, fatigue, mood swings, aggression, sleep disturbances, and elevated serotonin levels. An increase in GABA levels corresponded to increased fatigue, mood swings, aggression, sleep disturbances, and lack of concentration. Additionally, at a higher level of glutamate, statistically significant effects on the neuropsychic symptoms of the subjects were observed, as indicated by the Pearson r coefficient and a *p*-value < 0.01.

Furthermore, an elevated morning cortisol level was positively correlated with increased headaches and aggression. Conversely, a low DHEA level in the morning was associated with increased fatigue and greater lack of concentration, as demonstrated in Table 2. Additionally, a low DHEA level resulted in a significant increase in headache and aggression.

## 4. Discussion

Serotonin deficiency is associated with various symptoms, including sleep difficulties, eating disorders leading to weight gain, dyspepsia, lack of concentration, feelings of nervousness and worry, lack of drive, migraine, fibromyalgia, exhaustion, anxiety, and depression [31]. The possible causes of serotonin deficiency may include chronic inflammation, viral infections, and reduced nutrient absorption in the gut [32,33]. Therefore, it is advisable to undergo a gut health checkup to investigate serotonin deficiency. Additionally, it has been observed that a deficiency in tryptophan can be related to the malabsorption of lactose or fructose, leading to a serotonin deficiency [34]. In the current study, 81 patients exhibited serotonin levels outside the normal range, with the majority showing an excess.

A gender difference in serotonin secretion was observed in a 1997 study involving adult patients [35]. While this difference garnered considerable attention, it was not emphasized in a relevant manner until the subsequent study [36], which also considered other factors in children. Additionally, gender differences in serotonin levels could potentially impact SSRI treatment [37,38]. Notably, the patients in the current study were not undergoing allopathic treatment. Furthermore, gender-related variations in serotonin levels might also influence dietary choices [39], but further research in this area is warranted. It is worth mentioning that serotonin levels can vary with age and, for interpretation purposes, the obtained data were adjusted for age using categories (sublevel, normal, high), rather than solely relying on variations in serotonin levels.

Generally, a high-stress lifestyle, an unbalanced diet, and reduced physical activity can disrupt the hormonal and neurotransmitter balance, resulting in a serotonin deficiency [40,41].

A slight excess of serotonin can be caused, for example, by foods rich in tryptophan (such as fish, meat, nuts, and legumes) or by taking supplements containing tryptophan or Griffonia extract (as a source of 5-HTP) [42]. After approximately 4 weeks, a follow-up measurement of serotonin levels should be performed to assess whether the excess is permanent. If confirmed, further diagnostic and investigative measures should be considered in order to identify the underlying cause.

A high level of GABA primarily occurs as a counter-regulatory mechanism when excitatory neurotransmitters are excessively activated, such as in cases of increased stress [43]. This represents the body’s attempt to balance or at least mitigate the effects of increased stress hormones. Consequently, a normal level of catecholamines and an increased level of GABA may still indicate a high stress load [44]. In our study, 27 individuals exhibited excess GABA levels, with only one falling below the optimal range.

The level of GABA increases under anesthesia or following the administration of sedatives [45]. Therefore, an optimal balance between these two neurotransmitters is necessary for proper interaction. Additionally, interaction with serotonin, which enhances the effects of GABA, is important. Hence, a serotonin deficiency may limit the effectiveness of GABA [46].

An elevated level of glutamate can lead to symptoms such as anxiety, restlessness, hyperactivity, and muscle cramps [47]. In our study, 54 people had excessive glutamate levels, with none falling below the limit.

The effects of cortisol encompass various aspects, including stimulating metabolism and blood sugar to generate energy, regulating fat distribution in the body (with a tendency to accumulate fat in the abdominal area), increasing appetite, reducing pain sensitivity, altering emotional responses, inhibiting growth processes, lowering immunity, and suppressing inflammation [48]. On the other hand, continuous high cortisol levels can intensify inflammatory activity in the body [49]. In our study, the majority of patients had cortisol levels below the normal limit (81 people), with only 20% exhibiting levels above the normal limit at noon. In the evening, 27 people in this study were observed to experience an increase in cortisol levels.

A deficiency in DHEA can lead to a reduced ability to cope with stress [50]. Moreover, it amplifies the effects of cortisol due to the lack of up-regulation. This deficiency can manifest as malaise, depression, memory loss, and learning difficulties [51].

In women, a DHEA deficiency can also result in an insufficiency of sex hormones, leading to symptoms like premenstrual syndrome (pain, irritability, and mood swings before menstruation) and menopausal problems [52]. Chronic stress can also cause a DHEA deficiency (similar to cortisol deficiency) [53]. In our study, 108 people exhibited disturbances in DHEA levels in the morning, with only 27 people falling within the normal range. In the evening, 54 patients remained outside the normal range.

The limitations of this study can be attributed to the small number of patients and the presence of gastrointestinal disorders. Another limitation is the age factor, which may lead to different results than those that would be found in the adult population. The presence of patients diagnosed with mental illnesses and undergoing treatment with SSRIs or SNRIs, which could potentially obscure the outcomes of diet therapy and psychobiological therapy, could be regarded as a constraint or limitation.

A strong point of the study is the correlation between neuropsychological problems, gastrointestinal problems, and neurotransmitter levels, which can lead to more effective regulation, particularly in children, ensuring healthy and harmonious development.

## 5. Conclusions

Psychobiotics showed a statistically significant impact on improving hyperactivity, reducing aggression, and enhancing concentration at the end of the research period.

An excess of serotonin was associated with symptoms such as headaches, fatigue, mood swings, aggression, and sleep disturbances.

Increased GABA levels were linked to fatigue, mood swings, aggression, sleep disturbances, and impaired concentration.

Higher glutamate levels had a statistically significant influence on the neuropsychiatric symptomatology of the subjects.

Elevated morning cortisol levels showed positive correlations with headaches and aggressive behavior.

Low morning DHEA levels were associated with excessive fatigue and increased difficulty concentrating.

## Figures and Tables

**Figure 1 biomedicines-11-02643-f001:**
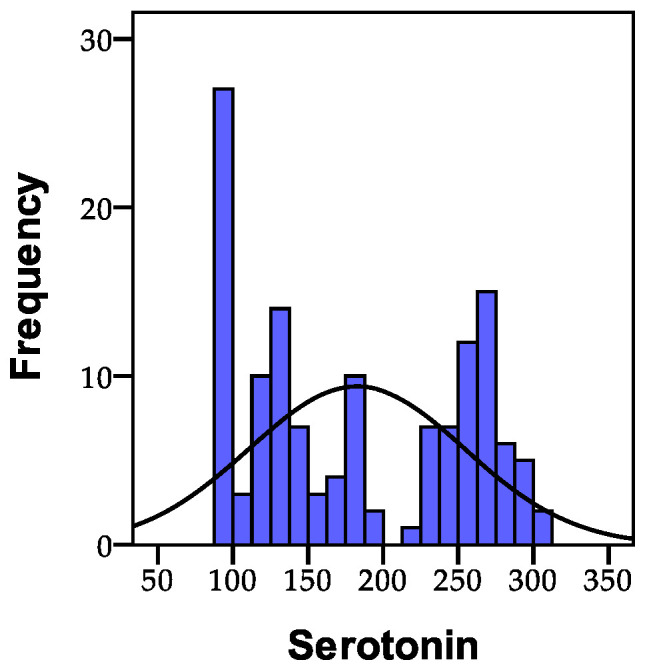
The distribution of the recorded serotonin values across the study cohort.

**Figure 2 biomedicines-11-02643-f002:**
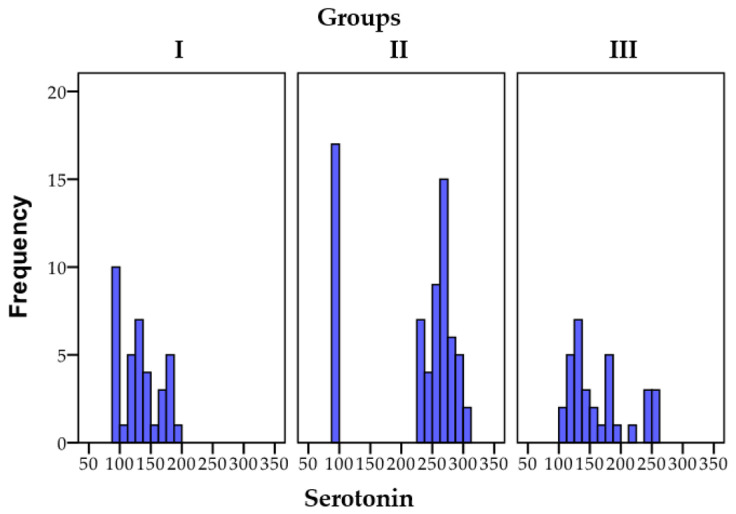
A graphical depiction of the final serotonin results corresponding to the three distinct groups.

**Figure 3 biomedicines-11-02643-f003:**
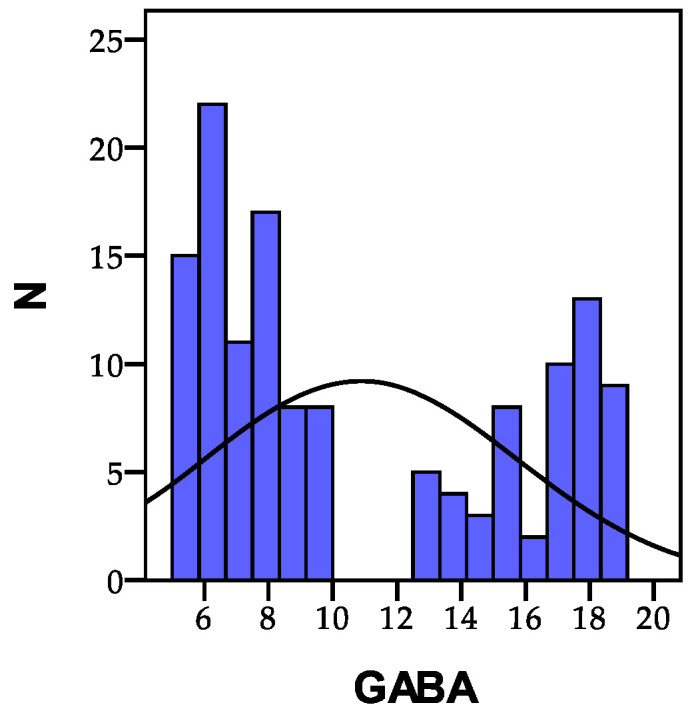
The distribution of the recorded GABA values within the study cohort; N = number of patients.

**Figure 4 biomedicines-11-02643-f004:**
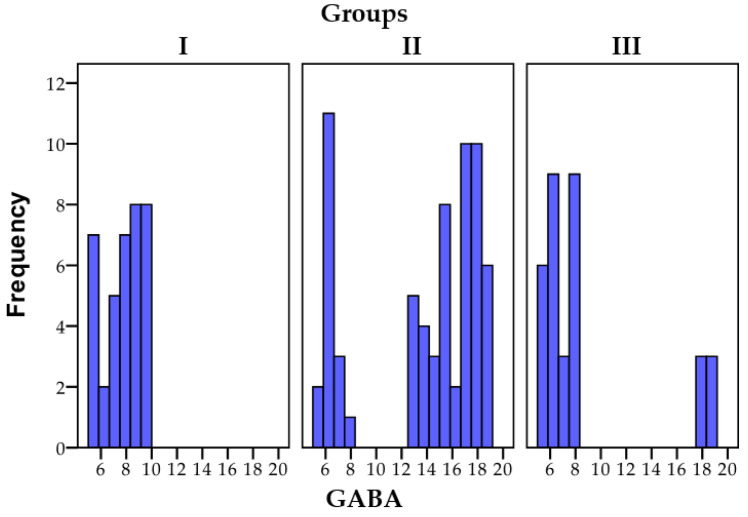
A graphical depiction of the GABA results across the three distinct groups; N = number of patients.

**Figure 5 biomedicines-11-02643-f005:**
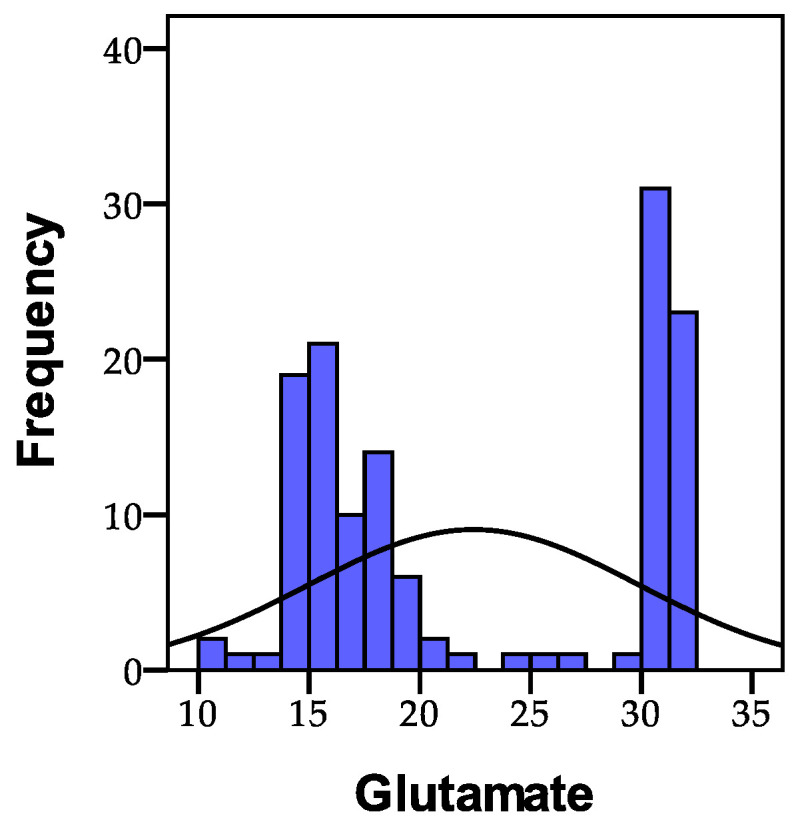
The glutamate value distribution across the study cohort; N = number of patients.

**Figure 6 biomedicines-11-02643-f006:**
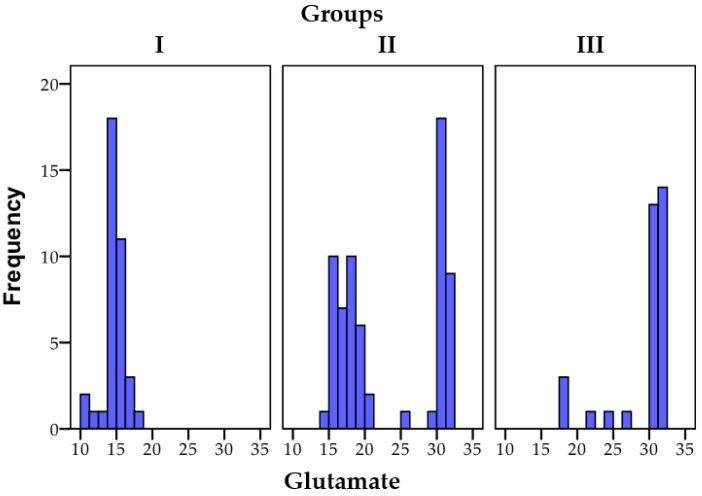
A visual depiction of the conclusive glutamate results across the three segregated groups; N = number of patients.

**Figure 7 biomedicines-11-02643-f007:**
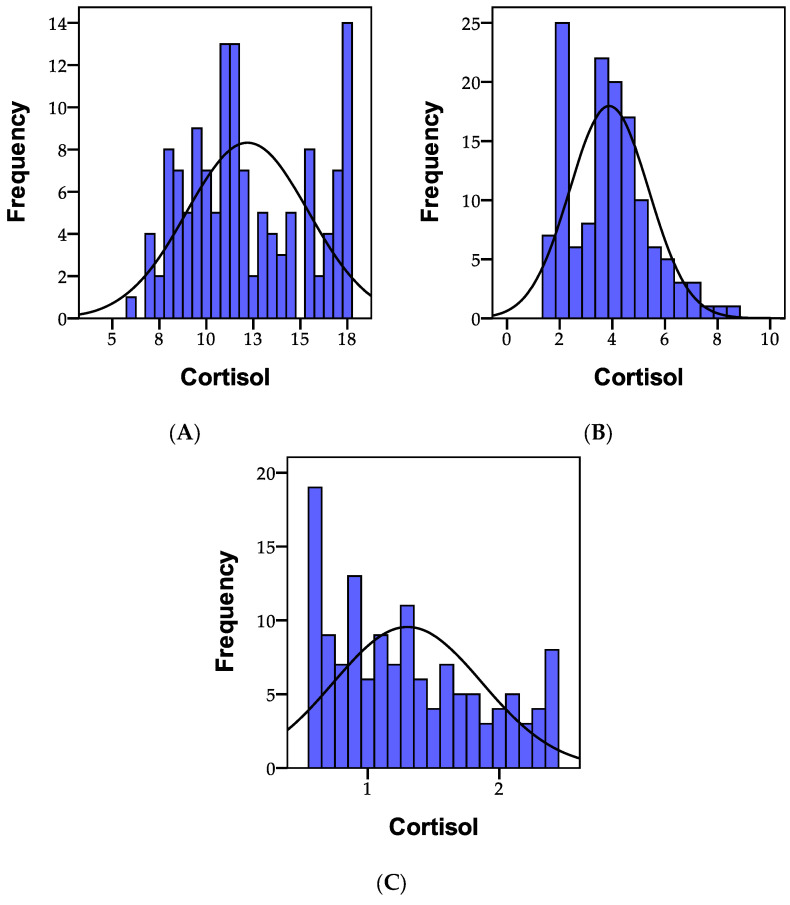
A representation of the recorded cortisol values in the morning (**A**), the afternoon (**B**) and the evening (**C**) within the study cohort; N = number of patients.

**Figure 8 biomedicines-11-02643-f008:**
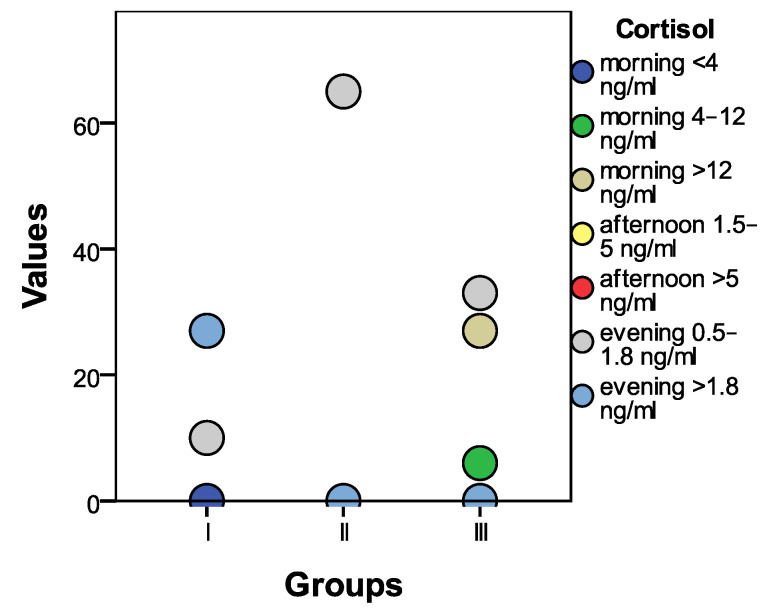
A graphical presentation of the conclusive cortisol results across the three distinct groups.

**Figure 9 biomedicines-11-02643-f009:**
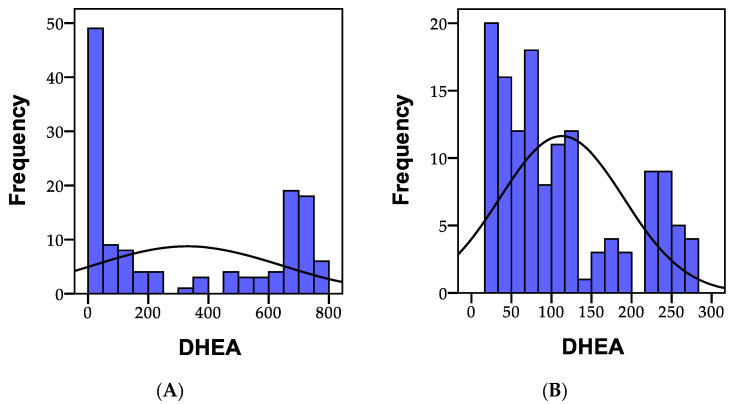
A representation of the recorded DHEA values in the morning (**A**), and in the evening (**B**) within the study cohort; N = number of patients.

**Figure 10 biomedicines-11-02643-f010:**
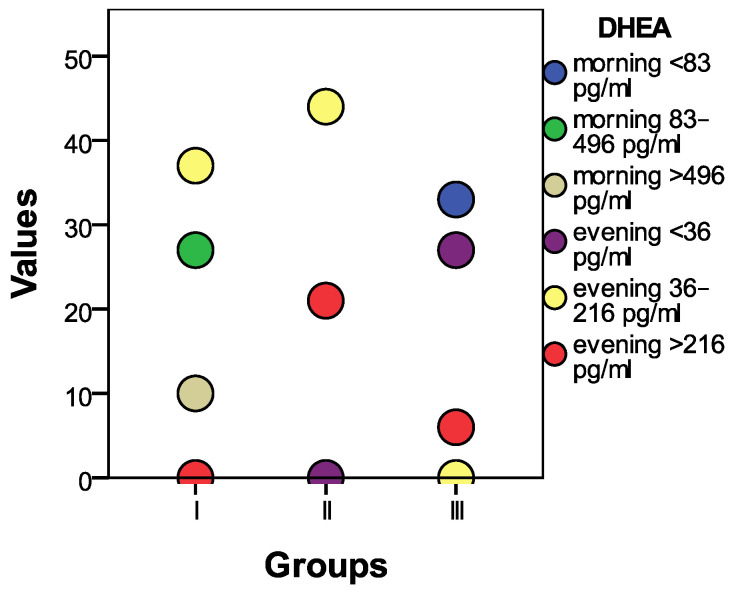
A graphical presentation of the conclusive DHEA results across the three distinct groups.

**Table 1 biomedicines-11-02643-t001:** Evaluation of neuropsychiatric manifestations according to gastrointestinal disorders.

		Groups						*p*	Total	
		I		II		III				
		N	%	N	%	N	%		N	%
Initial										
Headache	No	0	0.0	27	41.5	27	81.8	0.001 **	54	40.0
	Yes	37	100.0	38	58.5	6	18.2		81	60.0
Fatigue	No	0	0.0	48	73.8	33	100.0	0.001 **	81	60.0
	Yes	37	100.0	17	26.2	0	0.0		54	40.0
Mood swings	No	0	0.0	0	0.0	0	0.0	1.000 ^a^	0	0.0
	Yes	37	100.0	65	100.0	33	100.0		135	100.0
Hyperactivity	No	10	27.0	44	67.7	0	0.0	0.001 **	54	40.0
	Yes	27	73.0	21	32.3	33	100.0		81	60.0
Aggression	No	0	0.0	27	41.5	0	0.0	0.001 **	27	20.0
	Yes	37	100.0	38	58.5	33	100.0		108	80.0
Sleep disturbances	No	0	0.0	27	41.5	0	0.0	0.001 **	27	20.0
	Yes	37	100.0	38	58.5	33	100.0		108	80.0
Lack of concentration	No	27	73.0	0	0.0	0	0.0	0.001 **	27	20.0
	Yes	10	27.0	65	100.0	33	100.0		108	80.0
Final										
Headache	No	37	100.0	62	95.4	31	93.9	0.357	130	96.3
	Yes	0	0.0	3	4.6	2	6.1		5	3.7
Fatigue	No	35	94.6	65	100.0	33	100.0	0.068	133	98.5
	Yes	2	5.4	0	0.0	0	0.0		2	1.5
Mood swings	No	37	100.0	13	20.0	27	81.8	0.001 **	77	57.0
	Yes	0	0.0	52	80.0	6	18.2		58	43.0
Hyperactivity	No	10	27.0	60	92.3	31	93.9	0.001 **	101	74.8
	Yes	27	73.0	5	7.7	2	6.1		34	25.2
Aggression	No	37	100.0	65	100.0	0	0.0	1.000 ^a^	102	75.6
	Yes	0	0.0	0	0.0	33	100.0		33	24.4
Sleep disturbances	No	36	97.3	59	90.8	33	100.0	0.110	128	94.8
	Yes	1	2.7	6	9.2	0	0.0		7	5.2
Lack of concentration	No	37	100.0	44	67.7	27	81.8	0.001 **	108	80.0
	Yes	0	0.0	21	32.3	6	18.2		27	20.0

I = control group, II = group with psychoanxiety disorders, III = group with psychiatric disorders (III), *p* = statistically signification, ^a^ = cannot be compared, ** = correlation is significant at the 0.01 level.

**Table 2 biomedicines-11-02643-t002:** The Pearson correlation analysis pertaining to the connection between neurotransmitters and discrepancies in neuropsychiatric manifestations.

Pearson Correlation		Headache	Fatigue	Mood Swings	Hyperactivity	Aggression	Sleep Disturbances	Lack of Concentration
Serotonin	r	0.417 **	0.682 **	0.831 **	−0.068	0.426 **	0.709 **	0.162
	*p*	0.000	0.000	0.000	0.432	0.000	0.000	0.060
GABA	r	0.140	0.566 **	0.880 **	−0.064	0.169 *	0.569 **	0.363 **
	*p*	0.105	0.000	0.000	0.464	0.050	0.000	0.000
Glutamate	r	0.913 **	0.784 **	0.274 **	−0.356 **	0.921 **	0.469 **	−0.596 **
	*p*	0.000	0.000	0.001	0.000	0.000	0.000	0.000
Cortisol—morning	r	0.441 **	−0.009	−0.640 **	−0.225 **	0.418 **	−0.213 *	−0.272 **
	*p*	0.000	0.918	0.000	0.009	0.000	0.013	0.001
Cortisol—afternoon	r	−0.197 *	−0.122	−0.159	−0.136	−0.200 *	−0.197 *	0.771 **
	*p*	0.022	0.158	0.065	0.115	0.020	0.022	0.000
Cortisol—evening	r	0.059	−0.348 **	−0.591 **	0.145	0.040	−0.167	0.176 *
	*p*	0.496	0.000	0.000	0.093	0.642	0.053	0.041
DHEA—morning	r	0.075	−0.230 **	0.190 *	0.709 **	0.060	0.492 **	−0.563 **
	*p*	0.390	0.007	0.027	0.000	0.486	0.000	0.000
DHEA—evening	r	−0.597 **	−0.006	0.416 **	−0.078	−0.556 **	−0.185 *	0.569 **
	*p*	0.000	0.946	0.000	0.367	0.000	0.032	0.000
	N	135	135	135	135	135	135	135

r = Pearson coefficient, *p* = statistically significance, N = number of patients, * = correlation is significant at the 0.05 level, ** = correlation is significant at the 0.01 level.

## Data Availability

All the data processed in this article are part of the research for a doctoral thesis, which is archived in the aesthetic medical office where the interventions were performed.

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
