# Peer review of "Effect of Probiotic Therapy on Neuropsychiatric Manifestations in Children with Multiple Neurotransmitter Disorders: A Study"

_biomedicines, 2023, doi:10.3390/biomedicines11102643_

Round 1

Reviewer 1 Report

1. In the introduction section, the following sentence is repeated  "Significant amounts of vitamin B6 can be found in the following foods: whole grain products, potatoes, bananas, legumes (such as soybeans and lentils), avocados, carrots, Brussels sprouts, sunflower seeds, nuts, liver, meat, and fish"

2. Since the study decribes different groups of people, i.e. with psychoanxiety disorders and with psychiatric disorders, I was wondering what was the limitation of the study? If these people were diagnosed with such disorders, it is obvious that they possibly have undergone pharmacotherapy. Such specific drugs, such as SSRI, SNRI, alter neurotransmitter levels and thus may influence the results of the study; therefore making the results unreliable.

3. In line with this, the studied group consists of children in a range of 5-18, thus the neurotransmitter levels is not the same for every age and should be considered. Also, for instance, for serotonin levels it is known that there are differences in male and female subjects. Therefore, I suggest the Authors analyze the results considering age na sex also.

4. The Authors divided people into three groups, including the control. Since it is stated that all people had GI disorders and received personalized probiotic treatment, in my opinion also a control group without such a treatment should be included in order to show the normal level of compounds measured.

5. It is important to determine the type of the GI disorder as some of them require different treatment (i.e. antibiotics)

minor changes are required

Author Response

Reviewer 1

Firstly, we, the authors of the present manuscript wish to thank you for thoughtful commentary you have provided to improve the quality of the paper. We are very grateful for the time and effort you have devoted to this task. We have extensively revised my manuscript according to the recommendations. All changes in the text and the new figures that we have redesigned are highlighted. Please, see the point-by-point answers to your comments below. All correction was highlighted in the manuscript.

  1. In the introduction section, the following sentence is repeated "Significant amounts of vitamin B6 can be found in the following foods: whole grain products, potatoes, bananas, legumes (such as soybeans and lentils), avocados, carrots, Brussels sprouts, sunflower seeds, nuts, liver, meat, and fish"

Response: Thank you for observation. I deleted. (lines 56-58)

  1. Since the study decribes different groups of people, i.e. with psychoanxiety disorders and with psychiatric disorders, I was wondering what was the limitation of the study? If these people were diagnosed with such disorders, it is obvious that they possibly have undergone pharmacotherapy. Such specific drugs, such as SSRI, SNRI, alter neurotransmitter levels and thus may influence the results of the study; therefore making the results unreliable.

Response: Thank you for your comments. These individuals had imbalances and received natural treatment (Melissa extract, rhodiola, magnesium, vitamin B6) without SSRI or SNRI treatment in order to assess the effectiveness of probiotic therapy and diet therapy. I have noted the limitations as follows (lines 408-411):

Limitations: Patients diagnosed with mental illnesses who are currently receiving treatment with SSRIs or SNRIs, which have the potential to mask the outcomes of diet therapy and psychobiological therapy, are considered constraints or limitations.

  1. In line with this, the studied group consists of children in a range of 5-18, thus the neurotransmitter levels is not the same for every age and should be considered. Also, for instance, for serotonin levels it is known that there are differences in male and female subjects. Therefore, I suggest the Authors analyze the results considering age na sex also.

Response: Thank you very much for the comment. I completed the manuscript. (lines 355-364)

„ A gender difference in serotonin secretion was observed in a 1997 study involving adult patients [1]. While this difference garnered considerable attention, it wasn't emphasized in a relevant manner until the subsequent study [2], which also considered other factors in children. Additionally, gender differences in serotonin levels could potentially impact SSRI treatment [3,4]. Notably, the patients in the current study were not undergoing allopathic treatment. Furthermore, gender-related variations in serotonin levels might also influence dietary choices [5], but further research in this area is warranted. It's worth mentioning that serotonin levels can vary with age, and for interpretation purposes, the obtained data were adjusted for age using categories (sublevel, normal, high), rather than solely relying on variations in serotonin levels.”

  1. The Authors divided people into three groups, including the control. Since it is stated that all people had GI disorders and received personalized probiotic treatment, in my opinion also a control group without such a treatment should be included in order to show the normal level of compounds measured.

Response: Thank you for observation, we apologize for the confusion. We corrected the paragraph. (lines 130-135)

All patients had gastrointestinal disorders and adhered to healthy eating recommendations, which involved controlled caloric intake and set meal times. In the two study groups, they also received personalized probiotic treatment tailored to address their specific gastrointestinal issues. The recommended probiotics varied in terms of proportions and combinations, and they included bifidobacteria, lactobacilli, and saccharomyces ssp., with formulations that excluded gluten and dairy.

  1. It is important to determine the type of the GI disorder as some of them require different treatment (i.e. antibiotics)

Response: The selected patients, aged 5 to 18, had gastrointestinal disorders, as non-infectious diar-rhea, constipation, and other gastrointestinal disorders such as flatulence, feeling full, gas, belching, and abdominal pain. (lines 118,119)

Reviewer 2 Report

Introduction:

English is fine, but the content is diverse and not integrated. It sounds like a minireview.

Materials and methods:

“The recommended probiotics varied in proportions and combinations, including bifidobacteria, lactobacilli, and saccharomyces ssp., with compositions excluding gluten and dairy”

The recommended probiotics “varied”? I doubt the effects of probiotics on your patients with varied amount or content of probiotics. Please explain it.

Results

Figure 1 : Y-axis:” frequency” is not clear. How about “percentage” ? You also need to explain the meaning of percentage in your illustration.

Figure 2

The differences among group I, II and III are? Also the batch 1, 2, and 3?

Figure 8:in your Y axis: you show red circle and yellow circle, but there is no any circle in red and yellow in your plot?

Author Response

Reviewer 2

Firstly, we, the authors of the present manuscript wish to thank you for thoughtful commentary you have provided to improve the quality of the paper. We are very grateful for the time and effort you have devoted to this task. We have extensively revised my manuscript according to the recommendations. All changes in the text and the new figures that we have redesigned are highlighted. Please, see the point-by-point answers to your comments below. All correction was highlighted in the manuscript.

Introduction:

English is fine, but the content is diverse and not integrated. It sounds like a minireview.

 Materials and methods:

Comment 1: “The recommended probiotics varied in proportions and combinations, including bifidobacteria, lactobacilli, and saccharomyces ssp., with compositions excluding gluten and dairy”

 The recommended probiotics “varied”? I doubt the effects of probiotics on your patients with varied amount or content of probiotics. Please explain it.

Response 1: Again, we agree with your suggestion, and correction was made accordingly in the manuscript. (lines 133-136)

„The recommended probiotics varied in terms of proportions and combinations, and they included bifidobacteria, lactobacilli, and saccharomyces ssp., with formulations that excluded gluten, dairy, yeast, or egg. The dose of probiotics varied according to age and weight. In patients with constipation, it was supplemented with inulin.”

Results

Comment 2. Figure 1: Y-axis:” frequency” is not clear. How about “percentage”? You also need to explain the meaning of percentage in your illustration.

 Response 2: Thank you for amendment. Frequency was used instead of number of patient. I corrected accordingly.

Figure 2

Comment 3. The differences among group I, II and III are? Also the batch 1, 2, and 3?

 Response 3: Thank you for observation. It was a typo, both meaning the same thing. I corrected accordingly.

Comment 4: Figure 8:in your Y axis: you show red circle and yellow circle, but there is no any circle in red and yellow in your plot?

Response 4: Thank you for observation. The lack of circles is attributed to overlaps with other circles, as explained earlier in the figure (lines 273-284)

„Morning cortisol levels below 4 ng/ml were not recorded in any individual. Levels between 4-12 ng/ml were observed in 10 people from group I, 65 from group II, and 6 from group III. Cortisol levels exceeding 12 ng/ml were recorded in 27 individuals from groups I and III.

Afternoon cortisol levels below 1.5 ng/ml were not observed in any of the participants. Levels between 1.5-5 ng/ml were recorded in 10 people from group I, 65 people from group II, and 33 people from group III. Cortisol concentrations at lunch exceeding 5 ng/ml were recorded in 27 individuals from group I.

Evening cortisol levels below 0.5 ng/ml were not observed in any participant. Levels between 0.5-1.8 ng/ml were observed in 10 people from group I, 65 people from group II, and 33 people from group III. Cortisol levels exceeding 1.8 ng/ml in the evening were observed in 27 patients from group I. Graphical representation can be seen in Figure 8.”

Round 2

Reviewer 1 Report

The paper is improved greatly, thus i suggest its publication

English is fine now

Reviewer 2 Report

The authros have addressed most my questions. I recommend it to be accepted.